# Regulation of gene expression downstream of a novel Fgf/Erk pathway during Xenopus development

**Laura M. Cowell‡, Michael King‡, Helena West, Matthew Broadsmith, Paul Genever, Mary Elizabeth Pownall, Harry V. Isaacs\***

Department of Biology, University of York, Heslington, York, United Kingdom

‡ LMC and MK are contributed equally to this work and share first authorship.
\* harry.isaacs@york.ac.uk

**Data Availability Statement:** All relevant data are within the manuscript and its Supporting Information files. The raw RNA-seq files have been deposited in GEO. GenBank accession numbers:

## Abstract

Activation of Map kinase/Erk signalling downstream of fibroblast growth factor (Fgf) tyrosine kinase receptors regulates gene expression required for mesoderm induction and patterning of the anteroposterior axis during *Xenopus* development. We have proposed that a subset of Fgf target genes are activated in the embyo in response to inhibition of a transcriptional repressor. Here we investigate the hypothesis that Cic (Capicua), which was originally identified as a transcriptional repressor negatively regulated by receptor tyrosine kinase/Erk signalling in *Drosophila*, is involved in regulating Fgf target gene expression in *Xenopus*. We characterise *Xenopus Cic* and show that it is widely expressed in the embryo. Fgf overexpression or ectodermal wounding, both of which potently activate Erk, reduce Cic protein levels in embryonic cells. In keeping with our hypothesis, we show that Cic knockdown and Fgf overexpression have overlapping effects on embryo development and gene expression. Transcriptomic analysis identifies a cohort of genes that are up-regulated by Fgf overexpression and Cic knockdown. We investigate two of these genes as putative targets of the proposed Fgf/Erk/Cic axis: *fos* and *rasl11b*, which encode a leucine zipper transcription factor and a ras family GTPase, respectively. We identify Cic consensus binding sites in a highly conserved region of intron 1 in the *fos* gene and Cic sites in the upstream regions of several other Fgf/Cic co-regulated genes, including *rasl11b*. We show that expression of *fos* and *rasl11b* is blocked in the early mesoderm when Fgf and Erk signalling is inhibited. In addition, we show that *fos* and *rasl11b* expression is associated with the Fgf independent activation of Erk at the site of ectodermal wounding. Our data support a role for a Fgf/Erk/Cic axis in regulating a subset of Fgf target genes during gastrulation and is suggestive that Erk signalling is involved in regulating Cic target genes at the site of ectodermal wounding.

## Introduction

Fibroblast growth factors (Fgfs) are important regulators of gene expression during *Xenopus* development [1–4]. Binding of Fgf ligands to Fgf receptor tyrosine kinases (RTKs) activates

cic-l= OQ689245 cic-s= OQ689244 GEO series record= GSE228699.

**Funding:** MK was in receipt of BBSRC White Rose studentship BB/J014443/1. LC is in receipt of BBSRC White Rose Studentship BB/M011151/1. HVI is in receipt of a Leverhulme Emeritus Fellowship EM-2021-035.

**Competing interests:** The authors have declared that no competing interests exist.

several intracellular signal transduction pathways, including the Map kinase/Erk pathway [5–7]. Data indicate that Fgfs are the dominant activators of Erk signalling in the early *Xenopus* embryo [2, 8]. A number of transcription factors are involved in the regulation of Fgf/Erk target genes in *Xenopus*, including AP-1, a dimer of Fos and Jun proteins, and Ets family proteins [9, 10]. The prevailing model is that during Fgf signalling, Erk (either acting directly, or indirectly, via activation of additional target kinases) phosphorylates and regulates the activity of transcription factors, leading to changes in gene expression.

We have previously proposed that a subset of Fgf target genes are transcribed when Erk inhibits the activity of a labile transcriptional repressor [11, 12]. In the current study we investigate the Cic (Capicua) transcription factor as a candidate transcriptional repressor regulating gene expression downstream of Fgf signalling in the *Xenopus* embryo. Capicua was originally identified in *Drosophila* and studies of Capicua and its vertebrate orthologue Cic have shown that in the absence of EGFR signalling Cic binds to target genes and inhibits their transcription [13–15]. EGFR signalling activates ERK, which in turn phosphorylates and inhibits Cic resulting in de-repression of target genes [16–18]. While Cic has been proposed as a general mediator of RTK/Mapk signalling, reviewed [19], the role of FGFR as a potential upstream regulator of Cic mediated gene expression during early vertebrate development has not been investigated.

We hypothesise that in the absence of Fgf signalling, *Xenopus* Cic constitutively represses a subset of Fgf regulated genes but upon activation of Fgf/Erk signalling this repression is relieved. In the present study, we show that Cic is expressed early in *Xenopus* development and activation of Erk signalling destabilises Cic protein. Zygotic Cic knockdown causes a phenotype characterised by loss of anterior structures, similar to that reported for overexpression of Fgf4 [3, 20]. Our transcriptomic analysis demonstrates significant overlap of genes up-regulated by Cic knockdown and Fgf overexpression, including known targets of Fgf regulation and genes involved in regulating Fgf/Mapk signalling. We identify the *fos* and *rasl11* genes as putative targets of Cic. Both genes contain consensus Cic binding and are up-regulated by Cic knockdown and Fgf4 over-expression. Furthermore, *fos* and *rasl11b* are expressed dynamically in the mesoderm at gastrula stages and their normal expression requires Fgf/Erk signalling.

Wounding of the embryonic ectoderm also leads to a potent activation of Erk. We show that activation of expression from *fos* and *rasl11b* is a remarkably fast transcriptional response to wounding, and that the *fos* response to wounding is largely dependent on Erk signalling. Our data suggest a possible role at the site of embryonic wounding for Erk signalling in the regulation of gene expression via inhibition of the Cic transcriptional repressor.

## Materials and methods

### Ethics statement

Animal work was undertaken as part of the UK Home Office project licence to M.E.P. (POF245295) in accordance with the Animals Scientific Procedures Act (1986) and was approved by the Animal Welfare and Ethical Review Body at the University of York.

### Embryological methods

*Xenopus laevis* embryos were cultured in Normal Amphibian Medium/10 (NAM/10) and microinjection carried out in NAM/3 + 5% ficoll. *Xenopus tropicalis* embryos were cultured in Modified Ringers Solution/20 (MRS/20) and microinjection carried out in MRS/9 + 3% ficoll.

## Drug inhibitors

Drug treatments were carried out in 0.1% or 0.2% DMSO in MRS/20. Fgfr inhibitor SU5402 (Sigma-Aldrich) [21] and Mek1 and 2 inhibitor PD0325901 (Cell Guidance Systems) [22], treatments were at final concentrations of 200 μM and 25 μM, respectively.

## Tagged Cic proteins

A 4828 bp fragment from the Myc-CICf plasmid donated by the Huda Zoghbi lab [23] (Addgene plasmid #48185) and coding for N-terminal Myc epitope-tagged mouse CIC-S was cloned into the Stu1 and Xba1 site of pCs2+. Synthetic myc-Cic mRNA was synthesised using SP6 RNA polymerase [20]. PCR, using pcDNA5/FRT/TO GFP-Capicua plasmid [18], was used to generate a transcription template containing a functional SP6 RNA polymerase promoter, the bovine growth hormone polyA sequence and encoding an amino terminal fusion of eGFP to the *H. sapiens* CIC-S protein.

## Identification of Xenopus Cic sequences

The *Mus musculus* amino acid sequence was aligned to the *Xenopus tropicalis* genome to identify potential Cic protein coding exons. Primers were designed using the predicted exon sequence to amplify *Cic* cDNA fragments from *stage 25 Xenopus tropicalis cDNA* covering the full open reading frames of *cic-l* and *cic-s*. Overlapping fragments were used to produce consensus open reading frames and conceptual peptide sequences for Cic-l and Cic-s. The empirically determined sequences of *Xenopus tropicalis* cic-l and cic-s have been submitted to GenBank with accession numbers OQ689245 and OQ689244.

## RT-PCR and qPCR

Total RNA was isolated using TRI Reagent (Sigma)cDNA was synthesised with SuperScript II or IV Reverse Transcriptase from mRNA using either gene specific primers or random hexamers, for cloning of *Cic* cDNA fragments, or Oligo d(T) primers, for RT-PCR expression analysis. qPCR methods were as per [24]. Relative expression levels of each gene were calculated using the 2–ΔΔCt method and normalised to *dicer* [25].

## In situ hybridisation analysis

Analysis of mRNA expression was as per [26] with modifications described in [27]. Double in situ analysis was as described in [28]. However, for analysis of mRNA expression at wound sites, in order to avoid confounding tissue damage, membranes were not removed, and were permeabilised by pre-treating with 0.1M potassium dichromate in 5% acetic acid solution for 40 minutes, washing in PBS and then bleaching with 5% $H_2O_2$ under illumination for 45 minutes. After extensive washing, the standard protocol, without permeabilisation by proteinase K, was continued.

## Immunohistochemistry

Immunohistochemistry was undertaken as described in [29]. Primary antibodies were 1/10,000 dilution of anti-dpERK1+2 monoclonal mouse antibody (Sigma) and 1/5,000 dilution of anti-myc 9E-10a monoclonal mouse antibody (Cell Signalling Technology) at 4°C overnight. Secondary antibody was a 1/1000 dilution of horse anti-mouse IgG-alkaline phosphatse conjugated secondary (VectorLab). Immunostaining was visualised using BM Purple substrate (Roche).

## Western blot analysis

Embryos were homogenised in Phosphosafe buffer (Novagen) and used for western blotting. The primary antibodies were anti-myc mouse monoclonal 9E-10 (gift of Paul Pryor, University of York) at 1/500, β-catenin (Sigma) at 1/25,000, anti-dpERK mouse monoclonal (Sigma) at 1/5,000, total ERK rabbit polyclonal (Sigma) at 1/500,000 and anti-gfp mouse antibody (Clontech) at 1/4000. Secondary antibodies used were anti-mouse at 1/4,000 and anti-rabbit (Cell signalling Technology) at 1/2,000. Proteins were detected using BM chemiluminescence blotting substrate (Roche).

## TALEN targeting of Cic

TALENs were targeted to the HMG box of cic-s and cic-l in exon 6. The spacer region between the forward and reverse TALENs corresponds to the amino acid sequence (HRALV) in the HMG box of Cic-s and Cic-l. The forward TALEN targets the DNA sequence 5′ `TATGATCTTCAGCAAGCGG` 3‘ with RVD sequence `NG NI NG NN NI NG HD NG NG HD NI NN HD NI NI NN HD NN NN`. The reverse TALEN targets the DNA sequence 5′ `GGTTTGGGTGGCGCTGAT` 3′ with RVD sequence `NN NN NG NG NG NN NN NN NG NN NN HD NN HD NG NN NI NG`. TALENs were synthesised in the University of York, Biology Technology Facility.

## RNA-Seq analysis and bioinformatics

A 2100 bioanalyzer instrument (Agilent) was used for RNA quality control, and 9 Illumina next generation sequencing libraries created (water, Cska-Fgf4, and Cic-TALEN injected samples in triplicate). Samples were sequenced on the Illumina HiSeq 2000 platform with a coverage of > 40 million reads per sample. Initial data analysis, including aligning raw reads to the *Xenopus tropicalis* reference genome version 9.1, (https://www.ncbi.nlm.nih.gov/assembly/GCF_000004195.3/) [30] was undertaken in the University of York Technology Facility using Salmon [31]. Differential gene-level expression analysis was undertaken using Sleuth [32] and the counts aggregation method. The reduced model was defined as '~batch', and the full model '~batch + treatment' to mitigate variation accounted for by embryo batch. Both the FGF overexpression and CIC knockdown gene-level RNA-seq data was filtered to remove lowly expressed genes with average transcripts per million (TPM) $\geq 1.5$ in either the control or treatment data. Genes which met the criteria of *p*-value $\leq 0.01$ and effect size $\geq 1.75$ or $\leq 1/1.75$ (0.571) were defined as significantly up-regulated or down-regulated genes, respectively. The Integrative Genomics Viewer (IGV) (https://software.broadinstitute.org/software/igv/) [33] was used to produce figures of RNA-seq data aligned to the *Xenopus tropicalis* genome. FASTQ files and processed read data have been submitted to GEO (series record GSE228699).

Gene ontology analysis was carried out using the PANTHER Classification System (http://pantherdb.org/) [34]. Statistical over-representation for PANTHER GO-slim molecular function and biological process terms was determined using Fisher's exact test with calculation of false discovery rate (fdr). Enriched GO terms were filtered to fdr $\leq 0.05$ and fold-enrichment $\geq 2$.

Venn diagrams of gene overlaps were produced using GeneVenn (http://genevenn.sourceforge.net/) and significance of overlaps was calculated using the gene list overlap tool at http://nemates.org/MA/progs/representation.stats.html. Volcano plots were generated using the ggVolcanoR server at https://ggvolcanor.erc.monash.edu/ [35]. Heatmaps and cluster analysis of temporal expression patterns were produced using https://software.broadinstitute.org/morpheus. Temporal expression levels were derived from the RNA-seq dataset of clutch A ribosome depleted embryos (4 to 23 hours post-fertilisation) in [36]. During hierarchical clustering singleton clusters were removed. Multispecies genomic alignments were produced

using mLAGAN (http://lagan.stanford.edu/lagan_web/index.shtml) [37]. Curve parameters were: calculation window = 100 bp, Minimum conservation window = 100 bp and conservation identity cutoff = 70%. Motif scanning was undertaken using FIMO in the MEME suite using p< = 0.001 or p< = 0.0001 significance cutoffs (https://meme-suite.org/meme/tools/fimo) [38].

## Results

### Cic is highly conserved in Xenopus and is expressed in the embryo

Alternative splicing of exons 1 and 2 of *Xenopus cic* gives rise to two transcripts encoding peptides with different amino-termini, Cic-s (1593 amino acids, 169.3 kDa) and Cic-l (2514 amino acid protein (269.5 kDa) (S1A–S1D Fig). Overall the predicted sequence identity of both *Xenopus* proteins to their human CIC orthologues is 49%, however, the HMG box DNA binding domain is identical (S1C and S1D Fig). Other functional domains of CIC are also conserved in *Xenopus*, including the C1 repression domain, C2 Mapk docking domains and a previously identified monopartite nuclear localisation signal (S2C Fig) [39–41].

QPCR and in situ analysis of *Xenopus cic* expression show that there is maternal expression of *cic-l* and this is localised to the animal hemisphere (S2A and S2C Fig). Maternal expression of *cic-s* is low and not detected by in situ hybridization (S2B and S2C Fig). Zygotic expression of both *cic-l* and *cic-s* rises during late gastrula stages (S2A and S2B Fig) and at this stage expression is ubiquitous. Widespread expression of both isoforms continues into neurula stages, although higher levels of *cic-l* and *cic-s* are detected in the developing nervous system (S2C Fig). Low level, non-localised expression of *cic-l* and *cic-s* is maintained through tailbud stages, however, both are expressed at higher levels in the otic vesicle and branchial arches (S2C Fig).

### Activation of Erk signalling reduces Cic protein levels in embryonic cells

In *Drosophila*, activation of Erk signalling by the EGFR receptor tyrosine kinase causes degradation of CIC protein leading to de-repression of target genes (reviewed, [19]). We therefore investigated whether activating Erk signalling reduces levels of Cic protein in *Xenopus* embryonic cells. A Myc-epitope tagged Cic protein expresses efficiently from injected mRNA, and, in keeping with its role as a transcription factor, Cic-myc localises to the nucleus (Fig 1A). In gastrula stage embryos, activated diphospho-Erk (dpErk) is detected in a circumblastoporal ring corresponding to the region of endogenous Fgf signalling in the mesoderm (Fig 1B) [8]. At this stage in normal development dpErk is absent from the animal hemisphere (Fig 1C), however, localised Fgf4 over-expression or a puncture wound results in strong dpErk staining in the animal pole region (Fig 1D and 1E). Western blot analysis shows that activation of Fgf signalling or wounding reduces levels of Cic-myc protein in embryos relative to controls (Fig 1F).

The effects of overexpressing Cic on development were investigated using a gfp-CIC fusion protein, which is readily detected in *Xenopus* embryonic cells by western blot and its fluorescence can be monitored in live embryos (S3A–S3C Fig). As is the case with myc-Cic, gfp-CIC localises to the nucleus (S3D Fig). Overexpression has no obvious effects on embryo development (S3E and S3F Fig).

### Efficient TALEN mediated targeting of the Cic locus

Transcriptomic analysis at neurula stage 14 was undertaken to determine the effects from ectopic zygotic Fgf4 expression, driven from injected Cska-Fgf4 plasmid, or knockdown of

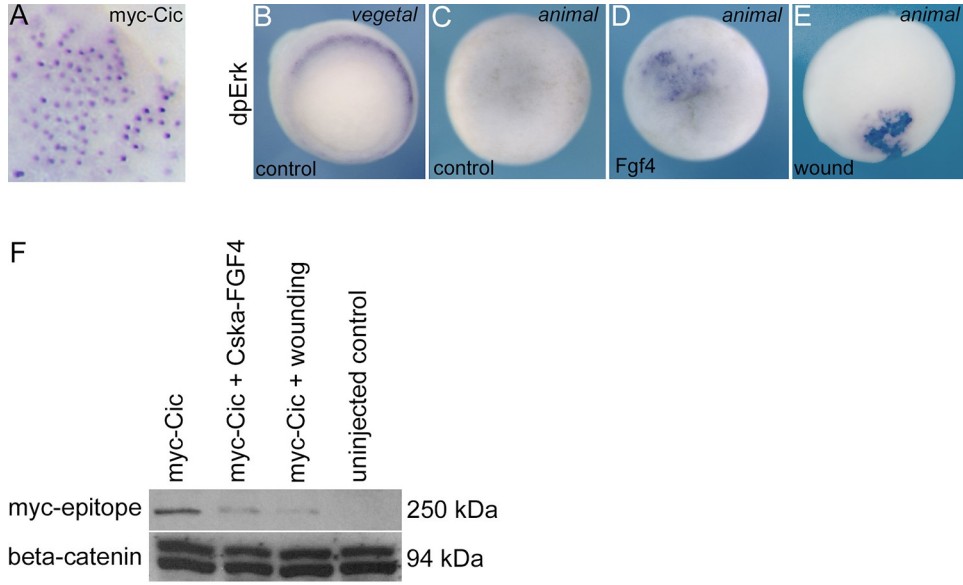

**Fig 1. Activation of Erk signalling reduces Cic protein in Xenopus embryonic cells. A,** Immunohistochemistry showing myc-tagged murine CIC-S expressed from injected synthetic mRNA (500 pg) localises to the nuclei of cells in the animal pole region of a *Xenopus laevis* embryo at gastrula stage 10.5. **B,** vegetal view of immunohistochemistry for activated diphospho Erk (dpErk) showing the staining in the nascent mesoderm around the blastopore in control embryos at gastrula stage 10.5. Staining is restricted to a known region of Fgf signalling. **C,** no dpErk is detected in the animal pole region of control embryos. **D**, localised injection of 10 pg Cska-Fgf plasmid or **E,** needle puncture wounding of the presumptive ectoderm results in ectopic Erk activation in cells of the animal hemisphere. **F,** Western blot showing that Cska-Fgf4 injection or puncture wounding reduces the levels of myc-Cic protein relative to control myc-Cic expressing gastrula stage 10.5 embryos. Embryos were collected 30 minutes after wounding. beta-Catenin is a loading control.

zygotic Cic function, mediated by expressing a transcription activator-like effector nuclease (TALEN) pair targeting the common HMG box encoding exon of *cic-s* and *cic-l* (Fig 2A and S1 Table). The rationale for targeting the DNA binding domain of Cic is that even in-frame INDELs are likely to disrupt DNA binding.

The sequence of PCR products amplified from the targeted exon in genomic DNA from individual F0 Cic-TALEN embryos matches wild-type sequence up to and including the forward TALEN target. However, downstream of this point sequence identity falls off, reflecting sequence heterogeneity arising from multiple different INDELs in F0 Cic-TALEN embryos (S4A Fig). Cloning of amplicon DNA and sequencing of single clones demonstrates the presence of a variety of INDELs, reflecting multiple independent TALEN targeting events in individual embryos (S4B Fig). RNA-seq data indicate *Cic* expression levels are very similar in control and Cic-TALEN embryos, (mean Transcripts Per Million (TPM) of 14.1 and 13.4, respectively, S1 Table). Mapping of individual sequencing reads to wild-type *Cic* sequence shows frequent mismatches in Cic-TALEN embryos, and a corresponding reduction of sequence coverage in the spacer region between the binding sites for the forward and reverse TALENs. We conclude that injection of the Cic-TALEN pair results in efficient, mosaic targeting of the Cic HMG box (S4C Fig).

## Axial defects in Cic knockdown embryos

In contrast to Cic overexpression that have no apparent phenotype, developmental defects are increased in embryos injected with the forward and reverse Cic-TALEN pair when compared to uninjected control embryos and embryos injected with only the forward Cic-TALEN (Fig 2B and 2C). A common feature of the Cic-knockdown phenotype is the loss of anterior

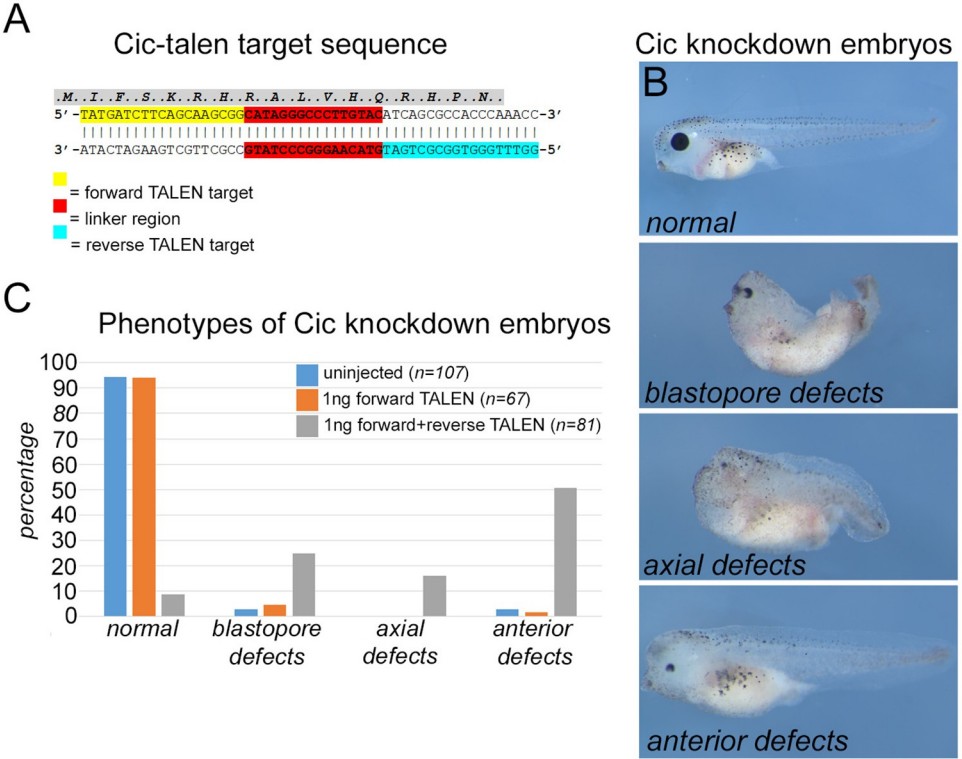

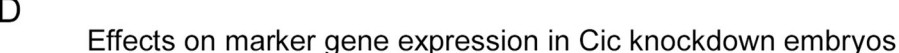

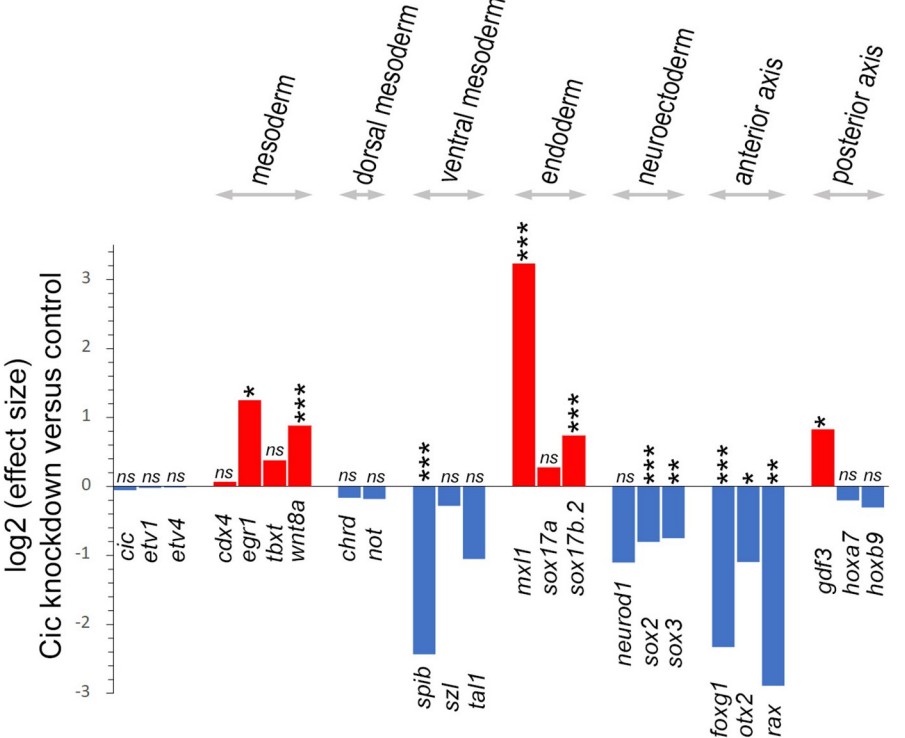

**Fig 2. Phenotypic effects of TALEN mediated Cic knockdown. A,** HMG box sequence in exon 6 of *Xenopus* cic gene targeted for the Cic-TALEN pair. Binding sites of the forward and reverse TALENs are indicated. The linker region

between the binding sites is targeted by the reformed bipartite endonuclease of the TALEN pair and subject to error prone non-homologous end joining repair. **B,** Stage 41 larval stage *Xenopus tropicalis* embryos showing phenotypes used to classify the effects of Cic-TALEN injection. *Anterior defect*s is mildest effect class characterised by loss of anterior structures, including cement gland and eye structures, while the rest of the axis relatively unaffected. *Axial defects* is an intermediate effect class that generally show loss of anterior structures and is accompanied by more general effects on axial elongation. The most severe class with *blastopore defects* exhibit varying degrees of blastopore non-closure with resulting failure of axial elongation and reduced anterior development. **C,** Bar chart showing classification of embryo phenotype in control uninjected, forward TALEN only and forward+reverse injected TALEN embryos. n values are combined from three separate experiments. **D**, changes in marker gene expression in embryos injected with the Cic-TALEN pair at neurula stage 14. $Log_2$ effect size and p values as determined by Sleuth analysis of RNA-seq data. Red bars indicated increased expression relative to control embryos. * p≤0.05, **p≤0.005 and *** p≤0.0005.

structures, including loss of the pigmented retina, which is sometimes accompanied by defects in axial extension and blastopore closure in more severely affected embryos. A similar range of phenotypes, including loss of anterior structures and failure to gastrulate, has been reported to result from Cska-Fgf4 overexpression [3]. It is worthwhile noting that failure to gastrulate also characterises the phenotype of *Xenopus* embryos where Fgf signalling is blocked. In this case aberrant morphogenesis results from loss of mesodermal gene expression [1, 3], whereas in Cska-Fgf4 embryos this likely results from ectopic mesodermal gene expression [3, 20].

Examination of a range of regional and tissue specific markers further characterises the Cic knockdown phenotype (Fig 2D and S1 Table). Interestingly, the levels *etv1* and *etv4* transcripts, which have previously been shown to be targets of Cic in mammals, are not significantly affected by Cic knockdown [18]. Expression of the mesodermal markers *egr1* and *wnt8a* are significantly upregulated, as are the endodermal markers *mix1l* and *sox17b* [2, 42–44], whereas expression of the mesodermal markers, *cdx4* and *tbxt* [20], are not significantly up-regulated. In keeping with the observed reduction in anterior neural development, there are significant reductions in expression of the anterior markers *foxg1*, *otx2* and *rax* [45–47]. The posterior marker gene *gdf3* (*derriere*) is increased but there are no significant effects on expression of the posterior markers *hoxa7* and *hoxb9* [20, 48]. The expression of *spib*, which is a marker of myeloid lineage development in the ventral mesoderm, is greatly down-regulated by Cic knockdown [49].

PANTHER gene ontology analysis of genes significantly affected by Cic knockdown (p≤0.01 and effect size +/- 1.75, S1 Table) reveals enrichment of molecular function terms associated with growth factor binding and DNA binding/transcriptional regulation (S2 Table). Biological process terms associated with signalling, gene transcription and neural/head development are also enriched (S3 Table).

## Fgf4 overexpression and Cic knockdown have similar effects on gene expression

A prediction of our hypothesis is that Fgf overexpression and Cic knockdown will have overlapping effects on gene expression. The scatterplot in Fig 3A supports this hypothesis and indicates that the genes significantly changing in Fgf4 overexpressing embryos show largely congruent up and down changes in Cic knockdown embryos.

## Genes involved in the Fgf/Mapk pathway are up-regulated by Fgf4 overexpression and Cic knockdown

Using the stated selection criteria, 122 genes are significantly up-regulated by Fgf4 overexpression and 457 by Cic knockdown (S1 Table; Fig 3A, 3B, 3D and 3E), with a highly significant (p<3.9e-63) overlap of 65 genes up-regulated in both conditions (Fig 3B, S4 Table).

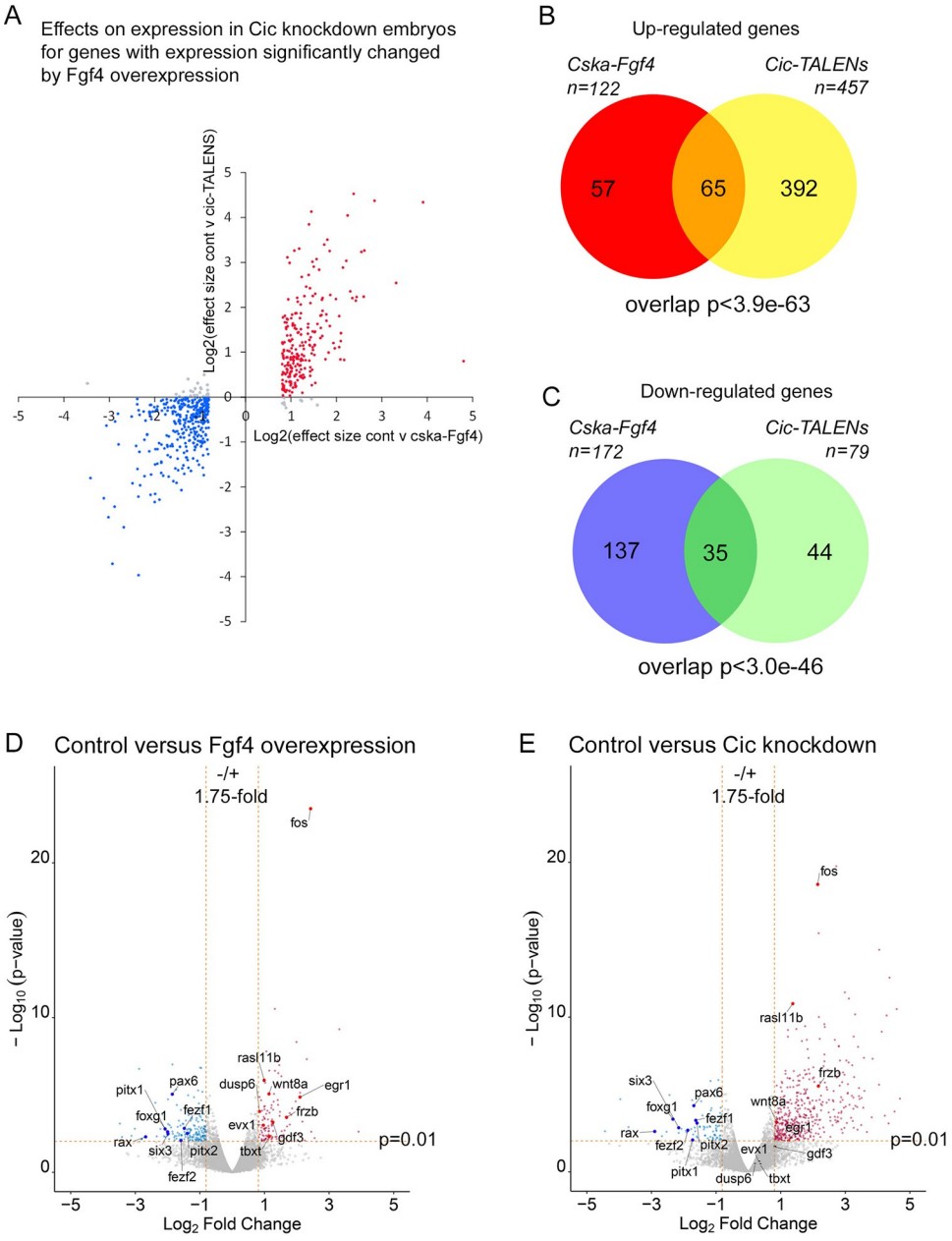

**Fig 3. Gene expression changes resulting from Fgf4 overexpression and Cic knockdown at early neurula stage 14.**
**A**, Scatterplot of changes in gene expression (log$_2$ effect size) at early neurula stage 14 following injection of 1 ng each of forward and reverse Cic-TALENs versus log$_2$ effect size in genes significantly changing in embryos injected with of 5 pg Cska-Fgf4 plasmid (p$\leq$ 0.01, effect size $\geq$1.75 or $\leq$0.571). Red and blue dots, respectively, indicate congruent up and down regulation. **B** and **C**, Venn diagrams of genes in *Xenopus tropicalis* embryos at early neurula stage 14 significantly up-regulated (p$\leq$0.01, effect size $\geq$1.75) or down-regulated (p$\leq$0.01, effect size $\leq$0.571) in Fgf4 overexpressing (Cska-Fgf4 injected) or Cic knockdown (Cic-TALEN injected) embryos, p-values of overlaps are indicated. **D** and **E**, Volcano plots of gene expression in control water injected embryos versus Fgf4 overexpressing embryos and control versus Cic knockdown embryos. Significantly up-regulated genes are indicated in red and down-regulated genes in blue. Grey points indicate genes below the significance thresholds.

The overlap gene cohort contains several previously identified targets of Fgf regulation, including *adamts1*, *egr1*, *frzb* and *wnt8* (S5 Table and references therein), while other known targets such as *dusp6*, *evx1*, *gdf3* and *tbxt* are significantly up-regulated by Fgf overexpression,

but not by Cic knockdown (Fig 3D and 3E). *fos* is the gene most dramatically up-regulated by Fgf4 overexpression (effect size 5.4) and Cic knockdown (effect size 4.4). QPCR was used to confirm that *fos* is up-regulated by Fgf4 and Cic knockdown (S5A Fig) and the well-established Fgf target gene *tbxt* is up-regulated by Fgf4 overexpression, but not by Cic knockdown (S5B Fig). We conclude that Cic is involved in the regulation of a subset of Fgf target genes.

## Genes involved in anterior development are down-regulated by Fgf4 overexpression and Cic knockdown

172 genes are down-regulated by Fgf4 overexpression and 79 by Cic knockdown embryos (S1 Table; Fig 3C, 3D and 3E). The observation that many more genes are up-regulated than down-regulated in Cic knockdown embryos is expected from inhibition of a transcriptional repressor (457 versus 79) (Fig 3B, 3C and 3E). Never-the-less, the overlap of 35 genes down-regulated in both conditions is significant (p<3.0e-46)(S6 Table). Many of the genes in the down-regulated cohort are known regulators of anterior axial development, including those for the Fezf1, Fezf2, Foxg1, Pax6, Rax and Six3 transcription factors (S7 Table and references therein).

## Fos and rasl11b are targets of transcriptional regulation by the Fgf/Erk pathway during gastrulation

In order to validate the cohort of genes up-regulated by Fgf4 overexpression and Cic knock-down as candidate targets of the putative Fgf/Erk/Cic pathway the regulation of two genes, *fos* and *rasl11b* were selected for further investigation. *Fos* is of considerable interest with regard to Fgf signalling because the Fos protein is a component of the AP-1 transcription factor, a known mediator of gene expression downstream of Fgf signalling in *Xenopus* [9]. While the AP-1 transcription factor is regulated by Erk kinase activity, this is the first report that Fgf regulates *fos* transcription during early *Xenopus* development. *Rasl11b* was selected because it encodes a divergent member of the ras family and, as such, *rasl11b* is a candidate component of a Fgf/Erk feedback pathway.

Cluster analysis of the temporal expression patterns of genes up-regulated by Fgf4 overexpression and Cic knockdown shows that *fos* and *rasl11b* both occupy cluster ii, where both are expressed most strongly during late blastula stages (Fig 4A). This differs from the expression of the known Fgf targets *frzb* and *wnt8a* [2], which are most highly expressed during gastrula stages (cluster i, Fig 4A).

Localised *fos* expression is first detected in the early gastrula embryo and in situ hybridisation analysis shows *fos* expression is detected in a circumblastoporal ring (Fig 4B and 4C) similar to the pattern of dpErk (Fig 4D). At this stage the *fos* expression domain appears somewhat larger than the dpErk domain, which may reflect differences in perdurance of dpErk protein, a transiently activated signal transduction intermediate, versus that of newly transcribed *fos* mRNA. *Fos* expression is dynamic, and as gastrulation proceeds, expression rapidly resolves to a domain above the dorsal blastopore lip (Fig 4E and 4F). *Fos* expression continues to overlap with dpErk, although the dpErk domain extends laterally and ventrally, forming a complete ring around the closing blastopore and persists in the involuted dorsal mesoderm as it extends along the developing axis (Fig 4G).

The regulation of *fos* expression by the Fgf/Erk pathway was investigated using the Fgf receptor drug inhibitor SU5402 and PD0325901, a drug inhibitor of Mek kinase, the upstream activator of Erk. Treatment of embryos with SU5402 or PD0325901 completely blocks Erk phosphorylation at the mid-gastrula stage and inhibits *fos* expression in gastrula stage embryos (Fig 4H–4O), indicating that Fgf signalling via Erk is required for *fos* expression in the gastrula.

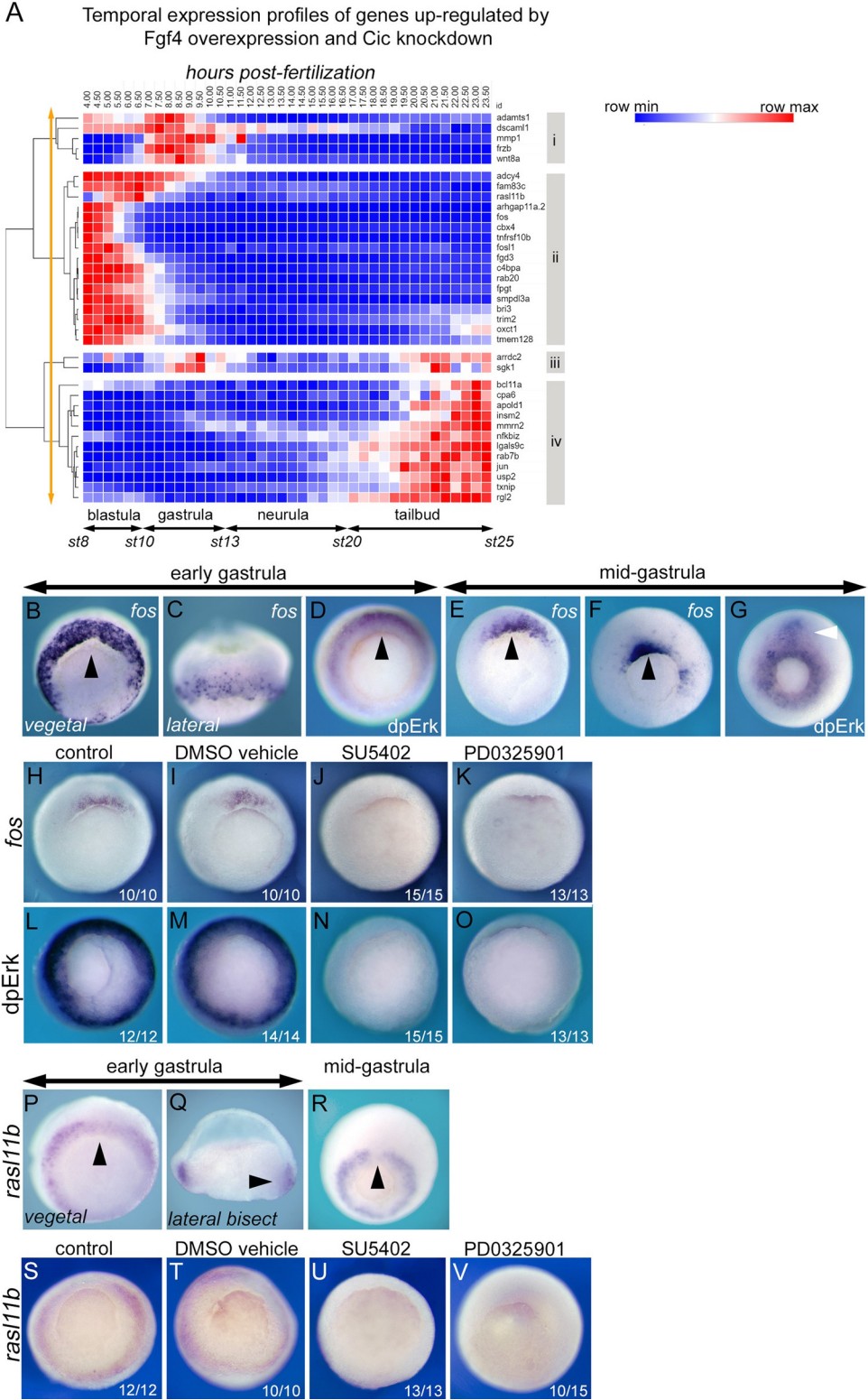

**Fig 4. Candidate targets of a Fgf/Erk/Cic pathway during gastrulation. A**, Cluster analysis of the temporal expression patterns (4 to 23 hours post-fertilisation at 24˚C) of genes up-regulated by both Fgf4 overexpression and Cic knockdown. Red shading indicates increased and blue shading decreased expression relative to the average expression level (white). Clusters i to iv are indicated. Orange arrow indicates the level at which the clustering tree was cut. **B, C, E** and **F,** are in situ hybridizations showing *fos* mRNA expression in early (stage 10+) and mid-gastrula (stage

10.5 and stage 11.5) *Xenopus tropicalis* embryos. **D** and **G** show immunohistochemical staining for dpErk in early (stage 10+) and mid-gastrula (stage11.5) embryos. Gastrula stage 10.5 embryos showing *fos* mRNA expression, **H-K**, or dpErk staining, **L-O**, in control embryos and embryos treated from stage 8 with 0.2% DMSO drug vehicle, 200μM SU5402 Fgfr inhibitor or 25 μM PD0325901 Mek inhibitor. **P-R** are in situ hybridizations showing *rasl11b* expression in early (stage 10+) and mid-gastrula (stage 11.5) embryos. **Q** is lateral view of a bisected embryo. **S-V** *rasl11b* expression in control embryos and embryos treated from stage 8 with 0.2% DMSO drug vehicle, 200μM SU5402 Fgfr inhibitor or 25 μM PD0325901 Mek inhibitor. All views are vegetal with dorsal to the top unless indicated. Black arrows indicate the dorsal blastopore lip. White arrow indicates involuted dorsal mesoderm. NB Indicated n values are from a representative experiment.

As with *fos*, *rasl11b* is expressed in a circumblastoporal domain in the early gastrula (Fig 4P and 4Q), overlapping the domain of Erk activity. However, the *rasl11b* domain in the posterior mesoderm remains more extensive than that of *fos* in later gastrula stages and, unlike *fos*, which is enriched in the dorso-lateral blastopore lip, *rasl11b* expression is reduced in this region (Fig 4R). Fig 4S–4V show that normal *rasl11b* expression is also dependent on Fgf/Erk signalling.

## Fos and rasl11b are not regulated by Fgf signalling pathway during later development

Localised *fos* expression is not detectable in the early neurula, but at later neurula stages *fos mRNA* and, as has been previously reported, dpErk are localised to the lateral edges of the neural plate as the neural folds move towards fusion in the dorsal midline (S6A–S6D Fig) [8]. SU5402 blocks Erk activation in known domains of Fgf activity, including the anterior neural ridge, forebrain, midbrain/hindbrain junction and posterior mesoderm, but has little effect on dpErk at the edge of the neural plate (S6E–S6M Fig). In contrast, PD0325901 blocks Erk activation in all domains, including the edge of the neural plate (S6E–S6G and S6N–S6P Fig). We conclude that Fgf receptor signalling is not a regulator of the Erk pathway at the lateral edge of the neural plate.

During late neurula and tailbud stages *rasl11b* is expressed in a segmental pattern in the presomitic mesoderm (S7A, S7B and S7G Fig). This pattern overlaps that of the *ripply2.2* (*bowline*) gene [50], which is also expressed in a segmental pattern in the presomitic mesoderm (S7C, S7D, S7H and S7I Fig). We note that during these stages the expression of domain of *rasl11b* is largely anterior to the region of Erk activity in the posterior mesoderm (S7E and S7F Fig). As noted before, drug inhibition of Fgf receptor signalling effectively blocks Erk activity in the posterior mesoderm but not in the neural folds, whereas inhibition of Mek blocks Erk activity in both domains (S7J–S7M Fig). S7N–S7Q Fig show that, in contrast to gastrula stages, expression of *rasl11b* in the presomitic mesoderm is refractory to treatment with SU5402 and PD0325901.

## Fos is a target of transcriptional regulation by Erk during wounding

It has been shown that wounding of embryonic ectoderm stimulates Erk phosphorylation and *fos* transcription [8, 51]. Here we investigate the relationship between Erk activation and *fos* expression. Fig 5A shows that at 5 minutes post-wounding there is strong activation of Erk that extends many cell diameters from the wound site in the flank of an early tailbud stage embryo. This dramatic Erk response rapidly subsides, so, by 15 minutes dpErk is restricted to a few cell diameters from the wound site and 60 minutes post-wounding is barely detectable. Fig 5B shows *fos* expression is activated rapidly by wounding but is delayed relative to Erk. Thus, strong *fos* expression is not detected until 15 minutes post-wounding, is still present at 60 minutes and has faded by 120 minutes. It has previously been reported that the Erk wound

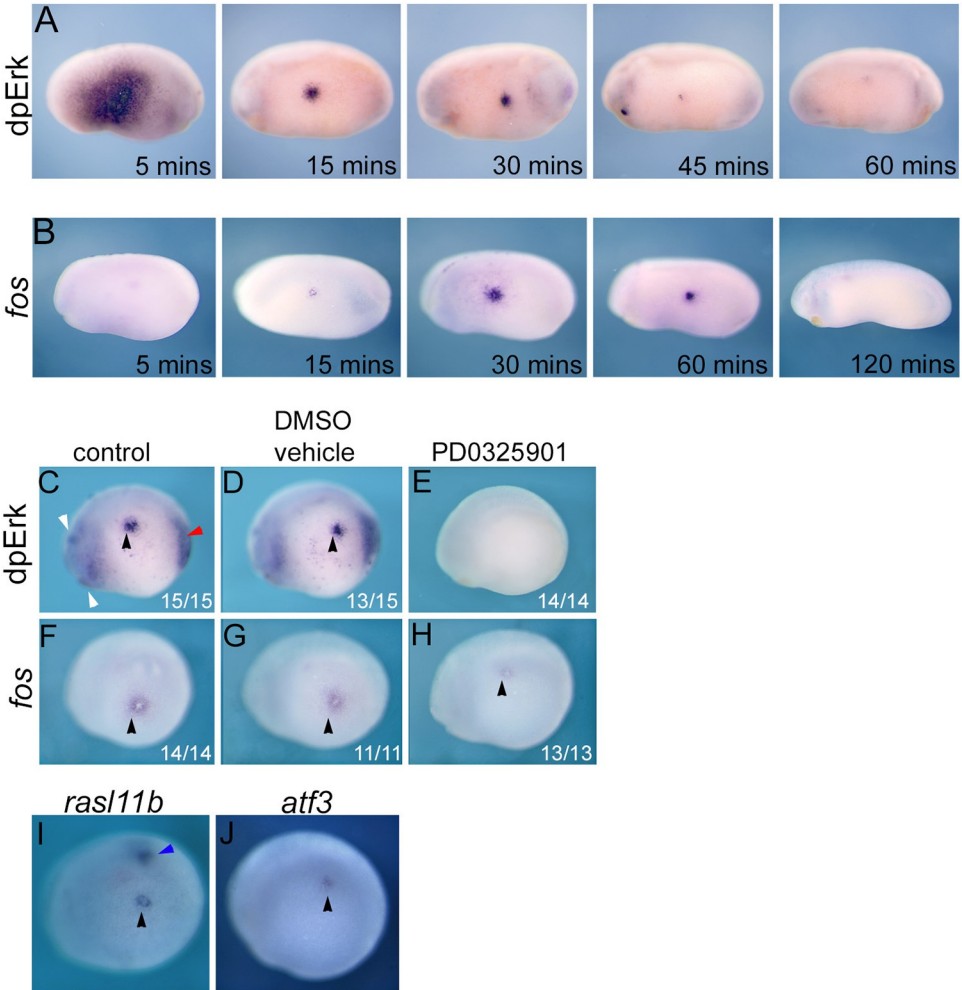

**Fig 5. Candidate targets of a Erk/Cic pathway during wounding. A,** time course (5 to 60 minutes) of dpErk expression following ectodermal puncture wounding on the flank of *Xenopus laevis* tailbud stage embryos. **B,** time course (5 to 120 minutes) of *fos* mRNA expression following wounding of *Xenopus laevis* tailbud stage 20 embryos. dpERK, **C-E,** and *fos*, **F-G,** expression at 30 minutes post-wounding in *Xenopus tropicalis* control embryos and embryos pre-treated with 0.1% DMSO drug vehicle or Mek inhibitor PD0325901 (25 μM in 0.1% DMSO) from neurula stage 14. **I,** activation of *rasl11b* expression at 10 minutes post-wounding. **J,** activation of *atf3* expression at 90 minutes post-wounding. Black arrow indicates site of wounding. White and red arrows indicate domains of dpErk corresponding to known regions of Fgf signalling in the anterior nervous system and posterior mesoderm, respectively. Blue arrow indicates endogenous *rasl11b* expression in the presomitic mesoderm. All are lateral views with anterior to the left and dorsal to the top. NB Indicated n values are from a representative experiment.

response in *Xenopus* does not require Fgf signalling [8]. Here we use PD0325901 to investigate whether Fgf independent Erk activation is required for the *fos* wound response. Fig 5C–5E show that the normal dpErk domains and wound induced Erk activation is inhibited by PD0325901. However, while Mek inhibition reduces *fos* expression, it does not completely eliminate it. We conclude that the Erk signalling is a regulator of *fos* transcription in response to embryonic wounding, although other signalling pathways might also be involved.

We investigated whether *rasl11b* and the *fos*-related gene *atf3*, which is also significantly up-regulated by Fgf4 overexpression (p≤0.001, effect size 1.76) and Cic knockdown (p≤0.0001, effect 1.96 (S1 and S4 Tables) are upregulated in response to wounding. Fig 5I shows that *rasl11b*, like *fos*, is rapidly activated by wounding. *atf3* is also activated by wounding

but this response is delayed compared to *fos* and *rasl11b* and is not detected until 90 minutes post-wounding (Fig 5J).

### Consensus Cic binding sites are present in a highly conserved region of fos intron 1

An attractive model for the rapid activation of gene transcription by Fgf signalling or wounding is that activated Erk de-represses Cic bound to genes already poised for transcription. Therefore, we investigated whether the *fos* gene sequence has sites for Cic repression. In order to identify conserved gene regulatory elements in the putative Cic target *fos*, a LAGAN alignment was performed using *Xenopus*, human, mouse and chicken *fos* genomic sequences (Fig 6A) [37]. Conserved regions outside coding exons were scanned for the CIC octameric consensus motif (T(G/C)AATG(A/G)/A) using the FIMO tool of the MEME suite [38]. A region of ~100 bp of sequence conservation was identified within 500 bp upstream of the transcription start site, however, no Cic sites were identified in this region. Of particular note is the presence of a highly conserved region of ~100 bp in intron 1 in human, mouse and chicken sharing 75% identity with *Xenopus* (Fig 6A and 6B). A scan of this region reveals the presence of two adjacent sites matching the Cic consensus in *Xenopus* and chicken, and one each in human and mouse (Fig 6B and 6C). This region also contains a highly conserved TPA

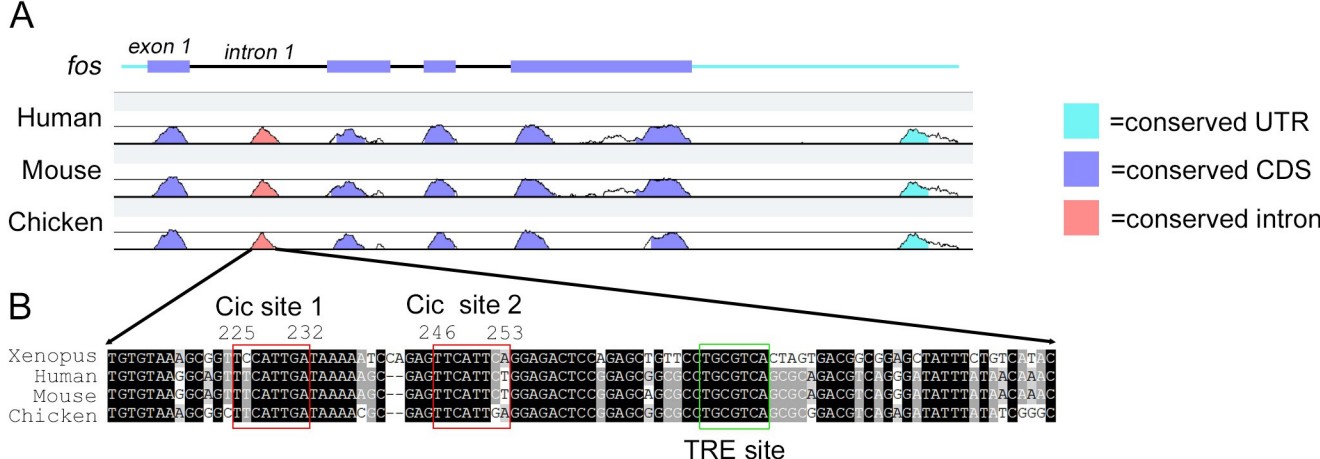

| Species | Intron 1 length (bp) | Sequence and position relative to start of intron 1 | Strand (+/-) | *p*-value | *q*-value |
|---|---|---|---|---|---|
| *X.tropicalis* (Frog) | 489 | 232 TCAATGGA 225<br>253 TGAATGAA 246 | -<br>- | 3.58e-05<br>7.93e-05 | 0.0636<br>0.0636 |
| *H.sapiens* (Human) | 753 | 427 TCAATGAA 420 | - | 7.93e-05 | 0.0636 |
| *M.musculus* (Mouse) | 754 | 422 TCAATGAA 415 | - | 7.93e-05 | 0.0636 |
| *G.gallus* (Chicken) | 438 | 242 TCAATGAA 236<br>262 TCAATGAA 255 | -<br>- | 7.93e-05<br>7.93e-05 | 0.0636<br>0.0636 |

**Fig 6. Conserved Cic binding sites are present in intron 1 of *fos*. A,** mLAGAN alignment of *fos* transcription units from *Xenopus tropicalis*, *Homo sapiens*, *Mus musculus* and *Gallus gallus* with *Xenopus* as the base sequence. Significant peaks of conservation are coloured as indicated. Figure is adapted from the VISTA-Point output (www.lagan.stanford.edu). **B,** Alignment of core sequences of *fos* intron 1 from *Xenopus*, human, mouse and chicken. Density of shading indicates level of conservation. Red boxes indicate sites of consensus Cic binding sites (T(G/C)AATG(A/G)A) in *Xenopus*. A conserved TPA responsive element (TRE) is indicated by the green box. **C,** table showing sequence, position, orientation and statistical significance of Cic consensus binding sites in *Xenopus*, human, mouse and chicken *fos* intron 1 identified by scanning with the FIMO tool from the MEME suite.

Responsive Element (TRE). The upstream region of several genes up-regulated by Fgf overexpression and Cic knockdown were also examined for potential Cic binding sites. S8 Fig shows the presence of sites closely matching the Cic consensus within 2kb of the transcription start of *fos*, *frzb*, *rasl11b* and *wnt8a*.

## Discussion

### Cic is a conserved regulator of RTK/Erk signalling

Previously, we have proposed that transcriptional de-repression is involved in the regulation of a subset of Fgf target genes [11]. In support of this we have shown that Fgf/Erk signalling modulates the activity of the Grg4 transcriptional repressor [12]. In the present study we investigate a potential role for the Capicua (Cic) transcriptional repressor in mediating Fgf/Erk signalling in early *Xenopus* development.

### Cic function in development and disease

Capicua was identified in *Drosophila* as a regulator of anteroposterior and dorsoventral patterning, acting downstream of the Torso and EGFR RTKs. In maternal Capicua mutant fly embryos, anterior and posterior development is maintained, but trunk and abdominal development is absent [13, 19]. In mice, loss of CIC in the forebrain promotes neural stem cell proliferation and the glial lineage, providing a link with frequently observed loss of CIC function in oligodendroglioma [52, 53]. Cic is also involved in the pathogenesis of the neurodegenerative disease spinocerebellar ataxia type I (SCA1) and a range of behavioural disorders through binding to mutant, disease-causing mutations of Ataxin-family proteins [54, 55]. Futhermore, CIC has been shown to be involved in lung and immune system development [56, 57].

Our hypothesis predicts that knockdown of the Cic repressor and Fgf overexpression will have similar effects on *Xenopus* development. Fgf signalling is required for posterior development of the main body axis, with Fgfs acting as dominant posteriorising signals that suppress anterior development [1, 3, 20, 58]. In support of our hypothesis, we show that Cic knockdown and Fgf overexpression down-regulate expression of a cohort of genes involved in anterior development and Cic inhibition causes anterior truncations similar to those caused by Fgf overexpression [20]. Given that Cic is a transcriptional repressor, down-regulation of anterior gene expression in response to its knockdown is unlikely to be direct, but rather reflects the involvement of Cic in regulating expression of genes involved posteriorizing the developing main axis. We note that *gdf3* (*derriere*), which is upregulated by Fgf4 overexpression and Cic knockdown, inhibits anterior development when ectopically expressed in the embryo [48].

### Mechanisms regulating Cic repression

Here we show that Fgf signalling or wounding, which potently activate Erk, both reduce Cic protein levels in *Xenopus* embryonic cells. The mechanisms regulating Cic transcriptional repression are complex. In *Drosophila*, activated ERK bound to the C2 domain phosphorylates CIC and inhibits transcriptional repression by context dependent mechanisms. During anteroposterior development activation of the Torso RTK by Trunk also leads to concentration-dependent CIC degradation, whereas in the ovary EGFR activation causes relocation of CIC from the nucleus to the cytoplasm [19].

In mammals a number of mechanisms have been proposed to regulate CIC mediated transcriptional repression. In EGF stimulated HEK293 cells, ERK phosphorylation of serine residues in the carboxy terminus of CIC inhibits binding of the nuclear importin protein KPNA3 to the nuclear localisation signal [18]. In addition, the p90rsk (RPS6KA1) kinase, which is

activated by ERK, phosphorylates serine[173], functions as a binding site for 14-3-3 proteins modulating DNA binding of the adjacent HMG box [18]. Phosphorylation of serine[173] has also been implicated in ubiquitin mediated degradation of CIC in glioblastoma cells [59].

There is also evidence for an ERK independent mechanism of CIC regulation involving SRC phosphorylation of a C-terminal tyrosine resulting in nuclear export of CIC [60]. The Src family protein Laloo is activated downstream of Fgf signalling in *Xenopus* mesoderm induction and could regulate Cic activity in this context [61, 62]. We note that the serine and tyrosine phosphosites, as well as the nuclear localisation signal discussed above are conserved in *Xenopus* Cic (S1 Fig).

## Regulation of gene expression downstream of Cic

Binding of CIC to gene loci recruits histone deacetylases leading to transcriptional repression underpinning CIC function in development and tumorigenesis [63]. Targets of CIC regulation commonly identified in mammalian transcriptomic studies are the PEA3 sub-family of ETS transcription factors and DUSP Mapk phosphatases [40, 64–66]. Interestingly, our transcriptomic analyses show that early expression of these genes does not significantly change in Cic knockdown embryos. However, it is possible that *Xenopus* Cic regulates of PEA3 genes during later development.

Our analysis identifies a cohort of putative Fgf/Cic target genes that are upregulated by Fgf4 overexpression and Cic knockdown. In addition to a subset of known Fgf transcriptional targets, such as *adamts1*, *egr1*, *frzb* and *wnt8* [2, 67], the up-regulated overlap cohort contains multiple components of the Fgf/Mapk pathway, including *atf3*, *arhgap11a*, *fam83c*, *fos*, *fosl1*, *ier3*, *jun*, *mixl*, *rab20*, *rab7b*, *rasl11b* and *rgl2* (S3 Table and references therein).

## Fos and rasl11b are regulated by a Fgf/Erk/Cic pathway during gastrulation

*Fos and rasl11b*, which are both up-regulated by Fgf overexpression and Cic knockdown, were selected for further investigation. Both *fos* and *rasl11b* are expressed in the early mesoderm and their normal expression during gastrulation is shown to require Fgf/Erk signalling.

The fos protein acting as a heterodimer with Jun, forms the AP-1 transcription factor required for transcriptional responses downstream of Fgf signalling [9]. Possible mechanisms for Fgf signalling post-transcriptionally regulating AP-1 activity are via phosphorylation of Jun by Jnk kinase and Fos by Erk kinase [68, 69]. The classical view of *fos* is as an immediate early response gene rapidly transcribed when activated ERK phosphorylates the ELK1 transcription factor, which in complex with SRF binds to serum response elements to activate transcription [70]. A similar mechanism is indicated for regulation of the immediate early gene *egr1* in *Xenopus* [71].

Our data suggest another level of regulation for *fos* transcription in the *Xenopus* embryo, by which Fgf mediated inhibition of Cic repression allows *fos* transcription. In support of this our analysis of a highly conserved region of *fos* intron 1 identifies consensus Cic binding sites. Intron 1 has previously been shown to contain gene regulatory elements and a number of DNAase I hypersensitive sites [72–74]. Furthermore, ChIP-Seq analysis has identified CIC binding in this region of *fos* in human neural stem cells and binding increases in the presence of a MEK inhibitor [63]. The intron 1 binding sites are upstream of a TPA responsive element (TRE) that has previously been shown to be capable of AP-1 binding [72], indicating further complexity to the feedback loops regulating *fos* expression. The AP-1 components *jun*, *fosl1* and *atf3* are also up-regulated by Cic knockdown and it is likely that after release from Cic repression a feed forward loop of AP-1 components auto-regulate their own expression and that of secondary response genes downstream of Fgf/Erk signalling.

Rasl11b, in common with other ras proteins, has GTPase activity, but unlike other ras proteins, rasl11b lacks lipid modifications and is not membrane tethered. A study in Zebrafish suggests that rasl11b modulates the activity of EGF-CFC protein One-Eyed-Pinhead (OEP)/Cripto, which is a co-receptor in the nodal signalling pathway. Interestingly, the *Xenopus* EGF-CFC proteins Frl1 and Frl2 have been implicated in Fgf signalling and were identified as Fgf receptor ligands, further suggesting a link between rasl11b and Fgf function. However, further studies will be required to elucidate any roles of rasl11b function in the Fgf signalling pathway.

### A possible role for Cic in regulating gene expression in the response to wounding?

Erk is rapidly activated in response to ectoderm wounding in *Xenopus* and this does not require Fgf signalling [8]. A role for Erk in formation of actomyosin structures required for wound closure, independent of transcriptional regulation, has been demonstrated [75]. However, while gene transcription is not required for the initial wound closure, complex patterns of gene expression are activated following *Xenopus* embryonic wounding and it has been proposed that these are involved in regulating tissue re-modelling post-wounding [51]. This latter study looked at changes in gene expression at the site of wounding in *Xenopus laevis*. S9A Fig shows that there is significant overlap between genes upregulated by Cic knockdown in *Xenopus tropicalis* and wounding in *Xenopus laevis*. In particular, we note that the putative Cic targets *fos*, *rasl11b* and *atf3*, which we show are up-regulated at the site of wounding in *Xenopus tropicalis*, are also up-regulated in response to wounding in *Xenopus laevis* (S9B Fig). Our demonstration that Cic protein is reduced by wounding and putative Cic targets genes are up-regulated at the wound site is suggestive that a Erk/Cic pathway also play a role in regulating the responses to wounding.

Erk regulated de-repression of Cic provides an attractive mechanism to rapidly activate transcription in response to Fgf signalling and wounding. A recent study indicates that activation of Erk leads to removal of *Drosophila* CIC from regulated enhancers and gene transcription within 5 minutes [76]. We note the similarly rapid response of *fos* and *rasl11b* transcription to wounding in *Xenopus*. This suggests that degradation or nuclear translocation of Cic protein is not required for the initial ERK mediated de-repression but is likely to be important for longer term stabilisation of Cic de-repression. Such rapid dynamics also provide a potential mechanism for re-imposing transcriptional repression of genes, such as *fos*, following transient Erk responses to growth factor signalling and wounding [8, 76, 77].

### Supporting information

**S1 Fig. Genomic organisation and sequence of *Xenopus tropicalis cic*. A,** diagram of the exon/intron structure of the *Xenopus cic* locus. **B,** diagram showing how alternative splicing of exons 1 and 2 give rise two Cic isoforms with different N-termini. **C,** Conceptual *Xenopus* Cic-s peptide sequence aligned with Cic-s from human and mouse. The conserved HMG box DNA binding domain, potential monopartite nuclear localisation signal, C1 repression domain, C2 Mapk docking domain, Mapk and Src phosphorylation sites are indicated. **D,** peptide sequence alignment of *Xenopus* and human Cic-l.
(JPG)

**S2 Fig. Temporal and spatial expression patterns of *Xenopus cic-s and cic-l*. A** and **B,** qPCR analysis of *cic-l* and *cic-s*. Expression is normalised to ubiquitously expressed control *dicer* and shown as relative to maximum expression level. **C,** in situ hybridisation analysis using isoform

specific antisense probes to *cic-s* and *cic-l*. Embryo orientations are indicated. *dbl* = dorsal blastopore lip, *ov* = otic vesicle and *ba* = branchial arches.
(JPG)

**S3 Fig. Developmental effects of gfp-CIC overexpression. A,** is a western blot detecting gfp
(~27 kDa) and gfp-tagged human CIC (~250 kDa) proteins translated in gastrula stage
embryos from injected mRNAs (image is uncropped). **B,** shows fluorescence of cytoplasmic
gfp protein in the animal hemisphere of a gastrula stage embryo. **C** and **D** (higher magnification), show strong nuclear expression of gfp-CIC. **E,** phenotype at larval stage 37 of uninjected
control embryos and **F,** embryos injected with 6ng synthetic gfp-CIC mRNA. NB Indicated n
values are from a representative experiment.
(JPG)

**S4 Fig. Analysis of gene targeting in Cic-TALEN knockdown embryos. A,** sequencing traces
of an amplicon from the region of *Cic* exon 6 targeted by the Cic TALEN pair using genomic
DNA from individual knockdown embryos as template. Wild-type sequence is included as a
comparison. Grey block indicates sequence bound by the Cic forward TALEN. **B,** sequence of
individual clones obtained from cloning exon 6 amplicon from an individual knockdown
embryo. Yellow shading indicates region of sequence mismatches with wild-type exon 6
sequence. **C,** individual RNA-seq reads from control water injected embryos and Cic-TALEN
injected embryos mapped to exon 6 of the *cic* locus. Combined BAM files from the three
experimental replicates were mapped to the *Xenopus tropicalis* genome using the IGV browser
and the output adapted. Dark grey indicates sequence mismatches. The spacer region between
the binding sites of the forward and reverse TALENs is indicated.
(JPG)

**S5 Fig. QPCR analysis of *fos* and *tbxt* expression. A** and **B**, qPCR analysis of *fos* and *tbxt*
mRNA expression in Fgf4 overexpressing, Cic knockdown and control water injected embryos
at neurula stage 14. Values are normalised to *dicer* mRNA expression and shown relative to
expression in control embryos.
(JPG)

**S6 Fig. Analysis of *fos* and activated Erk expression in the neural folds.** Dorsal views of neurula stage embryos showing localisation of *fos* mRNA by in situ hybridisation, **A** and **B,** and
dpErk by immunohistochemistry, **C** and **D**. **E-P**, immunolocalisation of dpErk in untreated
control embryos and embryos treated with 0.1% DMSO vehicle, 200 μM Fgfr inhibitor
SU5402 or 25 μM Mek inhibitor PD0325901. *anr* = anterior neural ridge, *bp* = blastopore, *fb* =
forebrain, *mhj* = midbrain/hindbrain border, *pm* = posterior mesoderm. NB Indicated n values are from a representative experiment.
(JPG)

**S7 Fig. Expression and regulation of *rasl11b* in the pre-somitic mesoderm. A** and **B**, in situ
hybridization of *rasl11b* expression at late neurula stage 19 and tailbud stage 22. **C** and **D**
expression of *ripply2.2* at late neurula stage 19 and tailbud stage 22. **E** and **F** localisation of
dpErk by immunohistochemistry at stage 19 and 22. **G-I** in situ hybridization analysis of
*rasl11b* (magenta) and *ripply2.2* (cyan) expression in the pre-somitic mesoderm at mid-tailbud
stage 25. **J-M**, neurula stage embryos showing immunolocalisation of dpErk and, **N-Q**, in situ
localisation of *rasl11b* expression in untreated control embryos and embryos treated with 0.1%
DMSO vehicle, 200 μM Fgfr inhibitor SU5402 or 25 μM Mek inhibitor PD0325901. Black
arrow indicates posterior mesoderm. NB Indicated n values are from a representative

experiment.
(JPG)

**S8 Fig. Identification of putative Cic binding sites in genes up-regulated by Fgf4 overexpression and Cic knockdown.** Position, orientation and statistical significance of Cic consensus binding sites in the 2kb upstream regions of putative targets of Cic regulation in *Xenopus*, identified by scanning with the FIMO tool from the MEME suite. Lower-case and upper-case sites are on the plus and minus strands, respectively.
(JPG)

**S9 Fig. Overlapping effects on gene expression in response to Cic knockdown in *Xenopus tropicalis* and ectodermal wounding in *Xenopus laevis*.** *Xenopus laevis* wounding datasets from [50] were filtered to only include genes annotated in the *Xenopus tropicalis* genome (12992 genes). Datasets for 30–90 minutes post-wounding were combined and filtered according to the following criteria, fold up-regulation $\geq 1.75$ and p-adj $\leq 0.1$. Duplicate gene entries were removed from the *Xenopus laevis* dataset. **A,** shows the highly significant overlap of genes up-regulated by Cic-knockdown in *Xenopus tropicalis* and wounding in X*enopus laevis*. **B,** Genes up-regulated by Cic-knockdown and ectodermal wounding. Orange shading indicates genes analysed at the site of wounding in this study (Fig 5).
(JPG)

**S10 Fig. Raw data for myc-epitope western blot in Fig 1D.**
(JPG)

**S11 Fig. Raw data for beta catenin western blot in Fig 1D.**
(JPG)

**S1 Table. Gene expression at neurula stage 14 in control, Fgf4 overexpressing and Cic knockdown embryos.**
(XLSX)

**S2 Table. PANTHER gene ontology slim molecular function analysis of genes up-regulated by Cic knockdown.**
(DOCX)

**S3 Table. PANTHER gene ontology biological process analysis of genes up-regulated by Cic knockdown.**
(DOCX)

**S4 Table. Genes significantly up-regulated by Fgf4 overexpression and Cic knockdown.**
(DOCX)

**S5 Table. Function of genes significantly up-regulated by Fgf4 overexpression and Cic knockdown.**
(DOCX)

**S6 Table. Genes significantly down-regulated by Fgf4 overexpression and Cic knockdown.**
(DOCX)

**S7 Table. Function of genes significantly down-regulated by Fgf4 overexpression and Cic knockdown.**
(DOCX)

## Acknowledgments

We would like to thank Sally James and John Davey of the University of York, Department of Biology, Technology Facility for RNA sequencing library construction and initial processing of the RNA-seq datasets.

## Author Contributions

**Conceptualization:** Harry V. Isaacs.

**Data curation:** Laura M. Cowell, Helena West, Harry V. Isaacs.

**Formal analysis:** Laura M. Cowell, Michael King, Harry V. Isaacs.

**Funding acquisition:** Harry V. Isaacs.

**Investigation:** Laura M. Cowell, Michael King, Helena West, Matthew Broadsmith, Harry V. Isaacs.

**Methodology:** Helena West.

**Project administration:** Mary Elizabeth Pownall, Harry V. Isaacs.

**Resources:** Mary Elizabeth Pownall.

**Supervision:** Paul Genever, Mary Elizabeth Pownall, Harry V. Isaacs.

**Visualization:** Michael King, Helena West, Matthew Broadsmith.

**Writing – original draft:** Helena West, Matthew Broadsmith, Mary Elizabeth Pownall, Harry V. Isaacs.

**Writing – review & editing:** Laura M. Cowell, Michael King, Matthew Broadsmith, Paul Genever, Harry V. Isaacs.

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
