## [Decision Letter · Decision Letter 0]

5 Jan 2023

PONE-D-22-31549The transcriptional repressor Cic is a mediator of Fgf/Erk signaling in early Xenopus developmentPLOS ONE

Dear Dr. Isaacs,

Thank you for submitting your manuscript to PLOS ONE. After careful consideration, we feel that it has merit but does not fully meet PLOS ONE’s publication criteria as it currently stands. Therefore, we invite you to submit a revised version of the manuscript that addresses the points raised during the review process.

Two experts have reviewed your manuscript and both find merit and interest in your work. Both raise a number of experiments which they believe will improve your manuscript. Please review their comments and address their concerns with experiments or a detailed response. 

Please submit your revised manuscript by Feb 19 2023 11:59PM. If you will need more time than this to complete your revisions, please reply to this message or contact the journal office at plosone@plos.org. Please include the following items when submitting your revised manuscript:

We look forward to receiving your revised manuscript.

Kind regards,

Edward Eivers

Academic Editor

PLOS ONE

Journal Requirements:

"No"

Reviewers' comments:

Reviewer's Responses to Questions

**Comments to the Author**

1. Is the manuscript technically sound, and do the data support the conclusions?

Reviewer #1: Partly

Reviewer #2: Yes

2. Has the statistical analysis been performed appropriately and rigorously? 

Reviewer #1: Yes

Reviewer #2: I Don't Know

3. Have the authors made all data underlying the findings in their manuscript fully available?

Reviewer #1: Yes

Reviewer #2: Yes

4. Is the manuscript presented in an intelligible fashion and written in standard English?

Reviewer #1: Yes

Reviewer #2: Yes

5. Review Comments to the Author

Reviewer #1: In this paper, the authors examined Xenopus fos gene as a Cic-regulated gene. Based on authors’ hypothesis that Cis downregulation and FGF overexpression showed similar effects on target gene expression, they performed RNAseq analysis and identified genes which is upregulated by both conditions. The authors next analyzed fos gene, one of them, and showed that the expression is seen in Xenopus gastrula in a pattern similar to dpErk activated domain. Furthermore, they found that fos was upregulated by wounding after massive activation of Erk.

The data provided by the authors are fundamentally interesting, but the study purpose is not clear because they first focused CIC but the in the later part they studied about fos, not CIC. In addition, CIC function for FGF signaling have already shown in Drosophila and mammals thus the role for the signaling in Xenopus is easily anticipated from these studies. If the authors want to clarify the function of CIC in Xenopus, the number of experiments is absolutely small (they showed only superficial phenotype in KD embryo). For accepting the paper, I recommend that the authors revise the question (and title) to match the data.

Major comments:

1. Although the title of the paper says “Cic is a mediator ….. in Xenopus development “, the authors showed only KD data, and the later part they carried out a series of experiments regarding fos. The authors should change the title to more appropriate one.

2. P3L58: If they authors aim to reveal that CIC constitutively inhibits target genes in the absence of FGF signaling, they should carry out other experiments (the analysis of target gene expression in CIC OE or KD for example)

3. Fig. 3: as commented above, the authors should add a series of experiment related to CIC function in Xenopus embryo. It is quite difficult to indicate the role of CIC from this data alone.

4. P8L219/Fig.3B: blastopore defect is also observed in FGF inhibited embryo (as described in author’s previous study). The author should discuss this point.

5. P8L242: Fos gene is well-known target of FGF signaling. If fos is chosen, RNAseq analysis using CIC makes little sense. The author should state the reason more clearly why they focused on fos gene (of course I understand that fos is the most differential gene by CIC-KD/FGF-OE). Readers will want to know other candidates. The author showed the result of rasl11b and atf3 in supplementary figure, but the data should move to main figure.

6. Fig. 4: The authors should confirm that CIC KD actually upregulates fos gene expression by in situ hybridization or RT-PCR.

7. P8L250/Fig. 4B and D: fos expression domain is larger than dpERK positive region. This result does not fully support the claim that fos is activated in the ERK pathway. How can this be interpreted?

8. P9L265-6, L267 and L269: the authors refer to the edge of the neural plate as “neural plate border”, but generally “neural plate border“ indicates the anterior region including neural crest and cranial placode. The description should be corrected.

9. Fig. S4K-M: The result that SU treatment did not abolish dpERK activation but PD completely inhibited dpERK activation is a little strange. Actually, the results shown in Fig.4N and O is opposite (the effect of SU treatment on dpErk activation is severer than PD) . The authors should at least explain this point.

Minor comments:

10. P2L29: neural plate plate margin

11. P20L677, L691 Fgf/CiC/Erk ->Fgf/Erk/Cic: Cic/Erk->Erk/Cic

12. SupFig1 legend is incorrect (D: Xt CIC-l and human CIC-l). Please check it.

13. Fig. S4N are missing.

Reviewer #2: Review Manuscript PONE-D-22-31549

Cowell and co-workers establish Cic (Capicua) as downstream target of Fgf/Erk signaling in the Xenopus embryo. Cic protein levels are reduced by Fgf signals and wounding-induced Erk activity. Using a TALEN targeting approach the authors show that knockdown of Cic causes gastrulation and primary axis defects. RNA-Seq analysis of embryos injected with FGF4 DNA and Cic-TALEN lead to the identification of common target genes, including the transcription factor fos. Bioinformatic analysis reveals a putative Cic binding site in a conserved intron 1 region of the fos gene. Exposure of early embryos to pharmacological drugs show that fos expression in the mesoderm depends on Fgf/Erk signaling and that wound-induced fos upregulation requires, at least in part, Erk activation. In addition, the transcriptomic analysis identifies ragl11b and atf3 as alternative targets of the Erk-Cic axis at the neural plate border and in response to ectodermal wounding, respectively. Together, Cic acts as a transcriptional repressor of several Fgf/Erk target genes in gastrulation, neural tube closure and wound healing.

The experiments are carefully carried out, the data are convincingly presented, and the manuscript is well written. Publication is recommended in PLOS-One, provided that the following comments are satisfactorily addressed:

1. The authors show nuclear localization of myc-Cic. Nuclear localization signals could be checked by bioinformatic analysis and, if detected, indicated in the Cic protein sequences.

2. The authors show that Cska-FGF4 and wounding decrease the protein levels of myc-Cic in mRNA-injected embryos. It would be interesting to see whether downregulation of FGF/Erk signaling stabilizes myc-Cic protein. A Westernblot analysis of myc-Cic-injected embryos after treatment with SU5402, PD0325901 or DMSO as control could be performed.

3. FGF/Erk signals induce mesoderm in Xenopus embryos. Does Cic knockdown promote mesodermal fate? Cic-TALEN and control embryos at stages 10-14 could be analyzed by whole-mount in situ-hybridization or RT-PCR analysis e.g. with the Xbra marker.

4. The authors show that knockdown of Cic causes blastopore closure, axial and anterior defects. How does overexpression of Cic affect Xenopus development? Can injection of myc-Cic rescue any of the Cic-TALEN phenotypes?

Minor comment:

Christen and Slack (1999. Development) already described dpErk activity at the neural plate border, for which credit should be given.

6. PLOS authors have the option to publish the peer review history of their article (what does this mean?). If published, this will include your full peer review and any attached files.

Reviewer #1: No

Reviewer #2: No

---

## [Author Response · Author response to Decision Letter 0]

13 Apr 2023

Dear Professor Eivers,

Many thanks for the opportunity to resubmit our manuscript [PONE-D-22-31549]. We found the referee comments very useful and address their specific comments in the annotated reviews below. This has entailed quite a lot of reorganisation and textual changes, which, as requested, are highlighted in the attached document. We feel that the changes have significantly improved the manuscript and we hope that you now find it suitable for publication in PLOS ONE.

As requested, we confirm that “The authors have declared that no competing interests exist.”

We look forward to hearing from you regarding this.

Relevant gel photos are now included in supporting data.

With very best regards,

Harry Isaacs (13-4-23)

 

Journal Requirements:

"No"

Response: Raw uncropped data for Figure 1D is now supplied in Figures_S18 and 18. There is a new western blot in Figure_S3A. Please note that this figure is uncropped and shows the whole of the blot. 

Response: Genbank accession numbers are now in the text. The RNA-seq data has been submitted to GEO and the series record accession number is now included. 

Response: This phrase is no longer used in the text.

Response: Caption text is included.

 

Reviewers' comments:

Reviewer's Responses to Questions

Comments to the Author

1. Is the manuscript technically sound, and do the data support the conclusions?

Reviewer #1: Partly

Reviewer #2: Yes

2. Has the statistical analysis been performed appropriately and rigorously?

Reviewer #1: Yes

Reviewer #2: I Don't Know

3. Have the authors made all data underlying the findings in their manuscript fully available?

Reviewer #1: Yes

Reviewer #2: Yes

4. Is the manuscript presented in an intelligible fashion and written in standard English?

Reviewer #1: Yes

Reviewer #2: Yes

5. Review Comments to the Author

Reviewer #1: In this paper, the authors examined Xenopus fos gene as a Cic-regulated gene. Based on authors’ hypothesis that Cis downregulation and FGF overexpression showed similar effects on target gene expression, they performed RNAseq analysis and identified genes which is upregulated by both conditions. The authors next analyzed fos gene, one of them, and showed that the expression is seen in Xenopus gastrula in a pattern similar to dpErk activated domain. Furthermore, they found that fos was upregulated by wounding after massive activation of Erk.

The data provided by the authors are fundamentally interesting, but the study purpose is not clear because they first focused CIC but the in the later part they studied about fos, not CIC. In addition, CIC function for FGF signaling have already shown in Drosophila and mammals thus the role for the signaling in Xenopus is easily anticipated from these studies. If the authors want to clarify the function of CIC in Xenopus, the number of experiments is absolutely small (they showed only superficial phenotype in KD embryo). For accepting the paper, I recommend that the authors revise the question (and title) to match the data.

Major comments:

1. Although the title of the paper says “Cic is a mediator ….. in Xenopus development “, the authors showed only KD data, and the later part they carried out a series of experiments regarding fos. The authors should change the title to more appropriate one.

Response: As requested we have re-formulated the title to :” Regulation of gene expression downstream of a novel Fgf/Erk pathway during Xenopus development”. This removes Cic from the title. We did not include fos in the title as it is used to validate conclusions about a much larger cohort of genes regulated in a similar manner.

2. P3L58: If they authors aim to reveal that CIC constitutively inhibits target genes in the absence of FGF signaling, they should carry out other experiments (the analysis of target gene expression in CIC OE or KD for example)

3. Fig. 3: as commented above, the authors should add a series of experiment related to CIC function in Xenopus embryo. It is quite difficult to indicate the role of CIC from this data alone.

Response to point 2 and 3: As requested we have now included a more detailed analysis of gene expression in Cic knockdown embryos using the RNA-seq data (Figure 2D), which is also supported by QPCR data (Figure_S10). We have also included PANTHER gene ontology analyses of targets of the Fgf/Erk/Cic pathway to reveal the types of processes regulated by this pathway (Tables_S6 and 7). These experiments provide a more detailed analysis of Cic function in Xenopus development, lines 253-269. We also include an analysis of Cic overexpression (see referee #2, response 4), lines 215-218.

4. P8L219/Fig.3B: blastopore defect is also observed in FGF inhibited embryo (as described in author’s previous study). The author should discuss this point.

Response: Yes, this is a good point and we now address this issue in the text, lines 247-252.

5. P8L242: Fos gene is well-known target of FGF signaling. If fos is chosen, RNAseq analysis using CIC makes little sense. The author should state the reason more clearly why they focused on fos gene (of course I understand that fos is the most differential gene by CIC-KD/FGF-OE). 

Response: Part of the interest of our observation is that Fos has not been shown to be transcriptionally regulated by Fgf signalling in Xenopus. We address the rationale for our interest more clearly in the text, lines 302-310. We have also included a new panel A in Figure 3 which shows clearly the congruent behaviour of a large cohort of genes in Cic knockdown and Fgf4 overexpressing embryos that provides additional support for our underpinning hypothesis that Fgf4 overexpression and Cic knockdown will have overlapping effects on development. 

6. Readers will want to know other candidates. The author showed the result of rasl11b and atf3 in supplementary figure, but the data should move to main figure.

6. Fig. 4: 

Response: Agreed, this is a good idea we have now moved the rasl11b normal expression into Figure 4 and have added new data regarding rasl11b expression by Fgf/Erk signalling (Figures 4P to V). We have also included the rasl11b and atf3 wounding data in Figures 5I and J. 

7. The authors should confirm that CIC KD actually upregulates fos gene expression by in situ hybridization or RT-PCR.

Response: QPCR data analysing fos expression are now included (Figure_S10), lines 284_289.

8. P8L250/Fig. 4B and D: fos expression domain is larger than dpERK positive region. This result does not fully support the claim that fos is activated in the ERK pathway. How can this be interpreted?

Response: The referee raises an interesting issue and this in now discussed in the text, lines 318-321.

9. P9L265-6, L267 and L269: the authors refer to the edge of the neural plate as “neural plate border”, but generally “neural plate border“ indicates the anterior region including neural crest and cranial placode. The description should be corrected.

Response: Good point, this could be confusing we address this in the text and now refer to “the edge of the neural plate”.

10. Fig. S4K-M: The result that SU treatment did not abolish dpERK activation but PD completely inhibited dpERK activation is a little strange. Actually, the results shown in Fig.4N and O is opposite (the effect of SU treatment on dpErk activation is severer than PD). The authors should at least explain this point.

Response: We address this in the text Line 346-348. The basic take home is that the Fgfr drug inhibitor has little effect on dpERK activity at the edge of the neural plate whereas the Mek inhibitor blocks it completely. This suggests that fos at the edge of the neural plate is likely to be regulated by Erk but this is independent of Fgf signalling, which is quite an interesting observation. 

Minor comments:

11. P2L29: neural plate plate margin

Response: As described above this is now addressed

12. P20L677, L691 Fgf/CiC/Erk ->Fgf/Erk/Cic: Cic/Erk->Erk/Cic

Response: Thanks for spotting this. We have now corrected.

13. SupFig1 legend is incorrect (D: Xt CIC-l and human CIC-l). Please check it.

Response: Now addressed

14. Fig. S4N are missing.

Response: Whoops, now corrected in the new figure.

Reviewer #2: Review Manuscript PONE-D-22-31549

Cowell and co-workers establish Cic (Capicua) as downstream target of Fgf/Erk signaling in the Xenopus embryo. Cic protein levels are reduced by Fgf signals and wounding-induced Erk activity. Using a TALEN targeting approach the authors show that knockdown of Cic causes gastrulation and primary axis defects. RNA-Seq analysis of embryos injected with FGF4 DNA and Cic-TALEN lead to the identification of common target genes, including the transcription factor fos. Bioinformatic analysis reveals a putative Cic binding site in a conserved intron 1 region of the fos gene. Exposure of early embryos to pharmacological drugs show that fos expression in the mesoderm depends on Fgf/Erk signaling and that wound-induced fos upregulation requires, at least in part, Erk activation. In addition, the transcriptomic analysis identifies ragl11b and atf3 as alternative targets of the Erk-Cic axis at the neural plate border and in response to ectodermal wounding, respectively. Together, Cic acts as a transcriptional repressor of several Fgf/Erk target genes in gastrulation, neural tube closure and wound healing.

The experiments are carefully carried out, the data are convincingly presented, and the manuscript is well written. Publication is recommended in PLOS-One, provided that the following comments are satisfactorily addressed:

1. The authors show nuclear localization of myc-Cic. Nuclear localization signals could be checked by bioinformatic analysis and, if detected, indicated in the Cic protein sequences.

Response: Now discussed in text and shown on Figure_S1, line 191 and lines 447-449.

2. The authors show that Cska-FGF4 and wounding decrease the protein levels of myc-Cic in mRNA-injected embryos. It would be interesting to see whether downregulation of FGF/Erk signaling stabilizes myc-Cic protein. A Western blot analysis of myc-Cic-injected embryos after treatment with SU5402, PD0325901 or DMSO as control could be performed.

Response: We agree that this might be interesting, but the referee does not indicate that this is essential before publication and feel that another experiment with overexpressed tagged protein would not add much to the current story. It would be even more interesting to carry out the suggested experiment and look for changes in the endogenous Cic protein. However, in the absence of a Xenopus Cic antibody we are unable to do this and as such, is beyond the scope of the current study.

3. FGF/Erk signals induce mesoderm in Xenopus embryos. Does Cic knockdown promote mesodermal fate? Cic-TALEN and control embryos at stages 10-14 could be analyzed by whole-mount in situ-hybridization or RT-PCR analysis e.g. with the Xbra marker.

Response: As requested we have now included a more detailed analysis of marker gene expression in Cic knockdown embryos using the RNA-seq data (Figure 2D), which is also supported by QPCR data (Figure_S10). This shows that there is little effect on the expression Xbra(Tbxt) suggesting that Cic knockdown does not promote mesodermal fate per se. 

4. The authors show that knockdown of Cic causes blastopore closure, axial and anterior defects. How does overexpression of Cic affect Xenopus development? Can injection of myc-Cic rescue any of the Cic-TALEN phenotypes?

Response: We now include a figure (Figure S4) showing that Cic overexpression has no obvious effects on development, lines 215-219. 

Minor comment:

Christen and Slack (1999. Development) already described dpErk activity at the neural plate border, for which credit should be given.

Response: Yes, an oversight on our part this is now addressed in the text.

6. PLOS authors have the option to publish the peer review history of their article (what does this mean?). If published, this will include your full peer review and any attached files.

Do you want your identity to be public for this peer review? For information about this choice, including consent withdrawal, please see our Privacy Policy.

Reviewer #1: No

Reviewer #2: No

---

## [Decision Letter · Decision Letter 1]

26 Apr 2023

PONE-D-22-31549R1 Regulation of gene expression downstream of a novel Fgf/Erk pathway during Xenopus development PLOS ONE

Dear Dr. Isaacs,

Thank you for submitting your manuscript to PLOS ONE. After careful consideration, we feel that it has merit but does not fully meet PLOS ONE’s publication criteria as it currently stands. Therefore, we invite you to submit a revised version of the manuscript that addresses the points raised during the review process.

Your manuscript has been reviewed by the same two initial reviewers. Both are happy with your answers to their comments and  ask for no addiitonal experiments. They only ask for you and the authors to address some minor comments. 

We look forward to receiving your revised manuscript.

Kind regards,

Edward Eivers

Academic Editor

PLOS ONE

Journal Requirements:

Reviewers' comments:

Reviewer's Responses to Questions

**Comments to the Author**

1. If the authors have adequately addressed your comments raised in a previous round of review and you feel that this manuscript is now acceptable for publication, you may indicate that here to bypass the “Comments to the Author” section, enter your conflict of interest statement in the “Confidential to Editor” section, and submit your "Accept" recommendation.

Reviewer #1: All comments have been addressed

Reviewer #2: (No Response)

2. Is the manuscript technically sound, and do the data support the conclusions?

Reviewer #1: No

Reviewer #2: Partly

3. Has the statistical analysis been performed appropriately and rigorously? 

Reviewer #1: Yes

Reviewer #2: I Don't Know

4. Have the authors made all data underlying the findings in their manuscript fully available?

Reviewer #1: Yes

Reviewer #2: Yes

5. Is the manuscript presented in an intelligible fashion and written in standard English?

Reviewer #1: Yes

Reviewer #2: Yes

6. Review Comments to the Author

Reviewer #1: The authors responded appropriately to the reviewers' comments. The addition of RT-PCR to Fig. 2D provides a more detailed description of what genes are affected by CiC KD. And results for other genes (rasl11, atf3) were added to the main figures, reducing the dissociation between the RNA-seq using CiC KD/Fgf4 OE embryos and the experiments in the later part. Other points have also been described in more detail. The revised manuscript was greatly improved.

Minor:

1)L376: In Fig. 3D/E, which dot shows atf3? It would be better to show it here.

Reviewer #2: The authors have now satisfactorily addressed all my comments! The new experimental data, including the demonstration that Cic knockdown has little effect on Xbra/tbxt expression (Fig. 2D, Fig. S10) and that gfp-Cic overexpression does not affect Xenopus development (Fig. S3) contribute to a more comprehensive understanding of the role of this protein. The identification and indication of a nuclear localization signal in the Cic protein (Fig. S1) is appreciated.

No further experiments are expected but the following points should be addressed before the manuscript can be accepted:

1. Lines 468-9. As far as I can see, no experimental evidence is given that supports the conclusion “Our data indicate that later expression domains of fos in the lateral edges of the neural plate … are not regulated by Fgf signalling.” Please correct!

2. Fig. 1F shows that wounding induces a decrease in Cic-myc protein levels. Please clarify how long the embryos were incubated after wounding before they were harvested for Westernblot analysis!

3. The new bar chart in in Fig. 2D helps to follow the Cip-Talen phenotype! For the sake of clarity, a subtitle for this panel such as “Cic-knockdown versus control” could be added within the figure. In addition, “Cic-Talen” should be indicated in panels A-C.

4. Line 352: Correct “the expression of domain of ras11b”!

5. Line 965: Figure 16 > Figure S16.

7. PLOS authors have the option to publish the peer review history of their article (what does this mean?). If published, this will include your full peer review and any attached files.

Reviewer #1: No

Reviewer #2: No

---

## [Author Response · Author response to Decision Letter 1]

2 May 2023

Reviewer #1: The authors responded appropriately to the reviewers' comments. The addition of RT-PCR to Fig. 2D provides a more detailed description of what genes are affected by CiC KD. And results for other genes (rasl11, atf3) were added to the main figures, reducing the dissociation between the RNA-seq using CiC KD/Fgf4 OE embryos and the experiments in the later part. Other points have also been described in more detail. The revised manuscript was greatly improved.

Minor:

1)L376: In Fig. 3D/E, which dot shows atf3? It would be better to show it here.

Response: This is now addressed in the text (See line 377).

Reviewer #2: The authors have now satisfactorily addressed all my comments! The new experimental data, including the demonstration that Cic knockdown has little effect on Xbra/tbxt expression (Fig. 2D, Fig. S10) and that gfp-Cic overexpression does not affect Xenopus development (Fig. S3) contribute to a more comprehensive understanding of the role of this protein. The identification and indication of a nuclear localization signal in the Cic protein (Fig. S1) is appreciated.

No further experiments are expected but the following points should be addressed before the manuscript can be accepted:

1. Lines 468-9. As far as I can see, no experimental evidence is given that supports the conclusion “Our data indicate that later expression domains of fos in the lateral edges of the neural plate … are not regulated by Fgf signalling.” Please correct!

Response: Sentence now deleted. Line 469

2. Fig. 1F shows that wounding induces a decrease in Cic-myc protein levels. Please clarify how long the embryos were incubated after wounding before they were harvested for Western blot analysis!

Response: Timing now included in figure legend. Line786-787

3. The new bar chart in in Fig. 2D helps to follow the Cip-Talen phenotype! For the sake of clarity, a subtitle for this panel such as “Cic-knockdown versus control” could be added within the figure. In addition, “Cic-Talen” should be indicated in panels A-C.

Response: Figure 2 now has subtitles for all panels

4. Line 352: Correct “the expression of domain of ras11b”!

Response: Now corrected

5. Line 965: Figure 16 > Figure S16.

Response: Now corrected. Line 966

---

## [Editor Report · Decision Letter 2]

9 May 2023

Regulation of gene expression downstream of a novel Fgf/Erk pathway during Xenopus development

PONE-D-22-31549R2

Dear Dr.Isaacs,

We’re pleased to inform you that your manuscript has been judged scientifically suitable for publication and will be formally accepted for publication once it meets all outstanding technical requirements.

Kind regards,

Edward Eivers

Academic Editor

PLOS ONE
---

## [Editor Report · Acceptance letter]

18 May 2023

PONE-D-22-31549R2 

Regulation of gene expression downstream of a novel Fgf/Erk pathway during Xenopus development 

Dear Dr. Isaacs:

I'm pleased to inform you that your manuscript has been deemed suitable for publication in PLOS ONE. Congratulations! Your manuscript is now with our production department. 

Kind regards, 

on behalf of

Dr. Edward Eivers 

Academic Editor

PLOS ONE